# Intestinal Production of Alpha-Glucosidase Inhibitor by *Bacillus coagulans* Spores

**DOI:** 10.3390/microorganisms11061462

**Published:** 2023-05-31

**Authors:** Hee-Woong Kim, Soo-Young Choi, Deug-Chan Lee, Hae-Ik Rhee

**Affiliations:** 1Department of Biomedical Technology, Kangwon National University, Kangwondaehakgil 1, Chuncheon 24341, Republic of Korea; 2College of Veterinary Medicine and Institute of Veterinary Science, Kangwon National University, Kangwondaehakgil 1, Chuncheon 24341, Republic of Korea; 3Institute of Bioscience and Biotechnology, Kangwon National University, Kangwondaehakgil 1, Chuncheon 24341, Republic of Korea; 4DALGIAL, K-Cube 101, Kangwon National University, Kangwondaehakgil 1, Chuncheon 24341, Republic of Korea

**Keywords:** α-gucosidase inhibitor, anti-lipidogenesis, *Bacillus coagulans*, facultative anaerobe, intestinal production

## Abstract

This study examines the possibility of directly producing and utilizing useful substances in the intestines of animals using anaerobic bacteria that can grow in the intestines of animals. A facultative anaerobe producing a large amount of α-glucosidase inhibitor was isolated from hay and identified and named *Bacillus coagulans* CC. The main compound of α-glucosidase inhibitor produced by *Bacillus coagulans* CC was identified as 1-deoxynojirimycin. α-glucosidase inhibitor activity was confirmed in the intestinal contents and feces of mice orally administered with spores of this strain, and it was confirmed that this strain could efficiently reach the intestines, proliferate, and produce α-glucosidase inhibitors. As a result of administering *Bacillus coagulans* CC to mice at 10^9^ cells per 1 kg body weight of spores for 8 weeks, the high-carbohydrate diet and the high-fat diet showed a 5% lower weight gain compared to the non-administrated group. At this point, in the spore-administered group, a decrease was observed in both the visceral and subcutaneous fat layers of the abdomen and thorax in both high-carbohydrate and high-fat diet groups compared to the non-administered group on computed tomography. The results of this study show that α-glucosidase inhibitors produced in the intestine by specific strains can work efficiently.

## 1. Introduction

The intestinal tract of animals is a micro-aerobic or anaerobic space and is a favorable environment for the growth of anaerobic or facultative anaerobic bacteria. Gut microbes affect the host by producing metabolites such as vitamins [1,2,3], organic acids [4,5], and enzymes [6,7,8] and affect the host [9,10,11,12]. To maximize the effect of intestinal microbes, it is common to consume various lactic acid bacteria in the form of food. However, lactic acid bacteria are vulnerable to acid and heat and have problems such as a low survival rate during product processing or after intake, so it is necessary to consider improving the efficient use of live bacteria [13,14,15,16]. Since spores produced by some bacteria have characteristics such as acid resistance, heat resistance, and chemical resistance, it has been reported that spores of anaerobic microorganisms reach the intestine with a high survival rate and then proliferate when orally administered [17]. In addition, when anaerobes such as *Bacillus licheniformis* produce secondary metabolites, the possibility of intestinal production of specific substances in the intestine has been suggested in a previous study [18].

Polysaccharides are decomposed to disaccharides, such as maltose and then decomposed to monosaccharides, such as glucose, by α-glucosidase in the intestine and absorbed into the bloodstream. The Inhibition of α-glucosidase during this process delays the decomposition of disaccharides and can control monosaccharide uptake. Since this process delays postprandial hyperglycemia caused by glucose intake, it is effective for diabetes [19], and there have been cases in which calorie intake was controlled by limiting glucose absorption in obesity [20,21].

As α-glucosidase inhibitors (AGI) of microbial origin, pseudooligosaccharide derived from *Actinoplanes* strains, 1-deoxynojirimycin, and tris base are known as secondary metabolites of microorganisms such as *Streptomyces* and *Bacillus* [22,23]. Recently, many studies have been conducted to test the efficacy of AGIs derived from natural products, such as mulberry leaf extract represented by 1-deoxynojirimycin or natto [21,22]. However, these primary processed products or extracts have the advantage of being mass-produced but have many limitations of practicality because they commonly have factors that reduce palatability, such as smell and color. In addition, the concentration of active ingredients is low, so there is a disadvantage that the desired purpose can be achieved only by continuously intake a certain amount per day [23,24,25,26].

In this study, we tried to suggest the possibility of in vivo production using anaerobic spores against the existing production method of physiologically active substances. For this purpose, spores of bacteria that produce secondary metabolites were orally administered to examine the production of specific substances in the intestine and their applicability to the body through proliferation in the intestine.

## 2. Materials and Methods

### 2.1. Strains 

Strains that grew in an anaerobic environment and produced AGI were isolated from domestically collected hay and silage. After suspending the collected sample in sterile physiological saline, heat treatment was performed at 80 °C for 30 min, and strains producing lactic acid were selected through enrichment culture from the supernatant [27]. This was anaerobically cultured on bromocresol purple (BCP, MB cell, MB-P1601) plate medium at 55 °C for 2 days to isolate and culture only yellow colonies, and then AGI activity was measured.

A soy flour medium was used as a medium for measuring AGI activity. After adding 5% soy flour to distilled water and inoculating the bacteria at a concentration of 10^6^ cells per 5 mL in a medium autoclaved at 121 °C for 15 min, the culture was incubated in an aerobic or anaerobic environment at 37 °C. After the end of the culture, the enzymes from bacteria in the culture medium were inactivated through an autoclave and AGI activity was measured using only the supernatant obtained by centrifugation (10,000× *g* rpm, 5 min).

### 2.2. Measurement of α-Glucosidase Inhibition Activity 

Alpha-Glucosidase was extracted and partially purified from the small intestine mucosa of porcine and used as an enzyme source [28,29]. And AGI activity was measured by Kim et al. (2007) method, and the inhibition rate was calculated by substituting it into the following equation [30].
The inhibition rate (%) = (A − B)/A × 100

In the above equation, A is the absorbance of the control group, B is the absorbance of the sample to which the inhibitor is added, and 1 AGI unit is defined as a unit of 1% inhibition of α-glucosidase used for measurement.

### 2.3. Identification of AGI 

AGI was confirmed by the comparative analysis of high-performance liquid chromatography (HPLC) and thin-layer chromatography (TLC) results using standard products. One milliliter of soybean flour medium culture solution was suspended, add 20 µL of 0.4 M potassium borate buffer (pH 8.5), and 20 µL of 5 mM 9-flurenylmethylchloroformate dissolved in acetonitrile, followed by a light-blocking reaction at room temperature for 30 min. Then, 20 µL of 0.1 M glycine to stop the reaction was added. The 1-deoxynojirimycin derivative prepared in the same way was analyzed and compared by HPLC. Briefly, an Agilent TC-C18 column (250 mm × 4.60 mm, 5 μm) was used for HPLC analysis; the mobile phase was acetonitrile/0.1% acetic acid (35:65, *v*/*v*). The flow rate was 1.2 mL min^−1^, and the column oven temperature was 40 °C and detected at 254 nm.

In the TLC, DNJ, and sample were spotted respectively on a high-performance thin layer chromatography (HPTLC, Supleco, 105516) plate, after development with propanol/acetic acid/water (4:1:1, *v*/*v*/*v*) solvent, the plate was scraped at 3 mm intervals and re-eluted with distilled water. The activity of the supernatant of the re-eluted sample was measured, and the migration distance was compared with DNJ.

### 2.4. Sporulation 

*Bacillus coagulans* sporulation was carried out in a two-step culture [31]. Incubate for at least 2 days in pre-culture medium (2% soytone, 1% yeast extract, 1% NaCl) until the number of bacteria reaches 10^8^ cells mL^−1^. Pre-culture solution at a concentration of 3% of the main culture medium (1% soytone, 0.5% yeast extract, 0.5% NaCl, 0.3% ammonium acetate, 0.1% MnSO_4_, 0.03% CaCl_2_, 0.01% MgSO_4_∙7H_2_O, FeSO_4_∙7H_2_O, and ZnSO_4_) inoculate and incubate for at least 3 days. Incubate until the number of spores in the main culture medium reaches at least 10^8^ cells mL^−1^. At this time, culture conditions were 37 °C, dissolved oxygen (DO) of 5 mg L^−1^ or more, and incubation was performed at a rotation speed of 360 rpm in the fermenter.

### 2.5. Animals

Experimental animals were bred in a designated research facility within Kangwon National University (Permit No. KW-190103-11). As experimental animals, 3-week-old ICR mice (KOATECH, Pyeongtaek, Republic of Korea) were used, and after the acclimatization period prior to the experiment, they were divided equally for 8 mice in each group and 4 mice per cage by body weight. The illuminance was controlled by lighting at intervals of 12 h, and the temperature of the breeding room was adjusted to 21 °C. Each group consisted of a standard group and a high-calorie feed group, including high-carbohydrate (HC) and high-fat (HF) feed. The protein, carbohydrate, and fat content of the feed was 55%, 22%, and 3.7% of the standard feed, and the HC feed was made up of 24%, 48%, and 4.8%, with 39%, 15.7%, and 26.1% for HF feed. The high-calorie feed group was divided into a spore-administered group and a non-administered group, respectively, and breeding was carried out. *B. coagulans* ATCC7050 was used as the strain to be used as a non-AGI-producing control in the spore-administered group. The spore-administered group was orally administered every other day by diluting the spores in 100 µL of physiological saline in an amount of 10^9^ spores per 1 kg of body weight. In the non-administered group, 100 µL of physiological saline was administered every other day instead of the spore dilution. During the breeding period, feed intake and body weight were measured at regular intervals, the fresh feces of the spore-administered group were collected, and the amount of bacteria and the activity of AGI activity were measured. After breeding, animals in each group were selected, anesthetized, and computed tomography (CT) scans of the thorax and abdomen were taken using a Toshiba Aelxion TSX-034A.

### 2.6. Statistical Analysis

The results of each experiment were expressed as the mean with standard error (±SE). A one-way analysis of variance (ANOVA) test (Bonferroni, SPSS, v.22, for Windows) was performed to determine the group means. Values were considered to be significant when *p* was less than 0.05 (*p* ≤ 0.05).

## 3. Results

### 3.1. Strain

Among the primary selection strains with high AGI activity, one strain identified as *B. coagulans* through 16s rRNA analysis with high activity and an excellent sporulation ability was finally selected. To improve the AGI production ability of this strain, N-methyl-N’-nitro-N-nitrosoguanidine (NTG, 100 µg mL^−1^) was treated to induce mutation until the mortality rate reached 99.9%. Among the mutants, a strain with improved AGI production ability was selected and named *Bacillus coagulans* CC (KCTC 14267BP).

As controls, *B. subtilis* DC-15 (KCTC 10763BP), an AGI-producing natto strain, and *B. coagulans* ATCC7050, a non-AGI-producing strain, were used. Table 1 shows the AGI activity generated by the aerobic or anaerobic culture at 37 °C. *B. coagulans* CC is a facultative anaerobic strain that grows under both aerobic and anaerobic conditions and produces AGI, but the growth rate and AGI productivity were high under aerobic conditions. The control strain, *B. subtilis* DC-15, is an aerobic strain used for natto production and produces a significant amount of AGI under aerobic conditions but grows poorly under anaerobic conditions and does not produce AGI at all. *B. coagulans* ATCC7050 was grown under both aerobic and anaerobic conditions but did not produce AGI. Therefore, *B. coagulans* CC is an AGI-producing facultative anaerobic strain, which is distinguished from other AGI-producing aerobic strains. On the other hand, *B. subtilis* DC15 produced the maximum AGI at 24 h of aerobic culture, whereas *B. coagulans* CC produced the maximum AGI at 48 h.

As a result of HPLC analysis of the cultures of *B. subtilis* DC-15 and *B. coagulans* CC, it was confirmed that 1-deoxynojirimycin (Sigma-Aldrich, D9305, St. Louis, MO, USA) showed peak and AGI activity at the same retention time (Figure 1). Separately, as a result of analyzing the culture medium by TLC, AGI activity was confirmed at the same migration distance as 1-deoxynojirimycin (DNJ). DNJ is an aza sugar-type AGI compound commonly found in various natural products, including those of microbial origin, and is reported to be the main AGI produced by microorganisms such as *Bacillus* and *Streptomyces* [32,33,34]. As a result of the above HPLC and HPTLC analysis of the culture of *B. coagulans* CC belonging to the genus *Bacillus*, the main compound of AGI was identified as DNJ. 

### 3.2. Production of AGI in Intestine by Spore Administration

To examine whether orally administered *B. coagulans* CC spores proliferate in the intestine and produce AGI, *B. coagulans* CC spores were administered, and the activity of AGI produced in the intestine was measured. During the diet, feces were collected regularly, and the amount of AGI excreted in the feces, and the number of *B. coagulans* CC were measured. Five milliliters of sterile water were added to 1 g of fresh feces, homogenized, and shaken at 4 °C for 15 min to obtain a suspension. A part of the suspension was autoclaved and centrifuged to measure the AGI activity in the supernatant. The remaining suspension was diluted with sterile water and cultured on a BCP agar medium, and the number of yellow colonies obtained was counted. Yellow colonies were observed in the feces suspension in the group administered with *B. coagulans* spores under the same culture conditions but not detected in the group not administered with the spores. Therefore, the yellow colony found in the feces of the spore-administrated group was regarded as *B. coagulans*.

As a result of analyzing the change in the number of bacteria excreted in the feces of the group administered with *B. coagulans* CC strain spores, it reached 10^7^ cells per 1 g of feces in the 4th week and was maintained. Therefore, it can be determined that the administered *B. coagulans* CC strain proliferates in the intestine, and some is excreted out of the body. On the other hand, after the cessation of spore administration in the 8th week of the diet, it was continuously detected until 2 weeks and then gradually decreased, indicating that the intestinal residence time of this strain was no more than several weeks (Figure 2).

The amount of AGI excreted in the feces of mice administered with *B. coagulans* CC spores was analyzed. As spore administration progressed, intestinal AGI increased and reached about 25,000 units per 1 g of feces in the 6th week, which is considered to be due to the saturation of *B. coagulans* CC intestinal proliferation (Figure 3). After the cessation of spore administration in the 8th week of the diet, fecal AGI decreased rapidly. Changes in the number of bacteria in feces and the amount of AGI according to spore administration were previously reported by Kim et al. (2022) and showed a similar trend [18]. In a previous study, *B. licheniformis* NY1505 also grew in the intestine and produced AGI, but the number and amount of AGI gradually decreased when spore administration was stopped because it did not stay in the intestine.

Subjects were sacrificed weekly during administration to confirm intestinal AGI production. The total weight of the intestinal tract was about 2 g, and the total protein amount of the AGI sample extracted from the whole intestinal tract was about 70 mg. Total AGI activity in the intestinal tract was 947 units in the 2nd week, and 2432 units and 6428 units in the 3rd and 4th weeks, respectively. On the other hand, the non-administered group did not show AGI activity in the intestinal tract, so it was confirmed that *B. coagulans* CC proliferated in the intestine after oral administration and produced AGI.

### 3.3. Spores Administration Effects against High-Calorie Diets

In addition to the previously reported AGI activity, DNJ has been reported to have many physiological functions [35,36]. However, in this study, the effect of DNJ produced in the intestine by spore administration on a high-calorie diet was approached from the perspective of weight gain. As a result of analyzing the effect of *B. coagulans* CC spore administration on the body weight of mice fed a high-calorie diet, *B. coagulans* CC spore-administered group showed a decrease in body weight compared to the non-administered group. In the 11th week of the HC diet, the body weight of the *B. coagulans* CC spore-administered group lost by about 5% compared to the non-administered group. In the case of the HF diet, *B. coagulans* CC spore-administered group lost about 7% of body weight compared to the non-administered group in the 11th week. As a control, *B. coagulans* ATCC7050 spore-administered group did not lose weight compared to the non-administered group in both HC and HF diets (Figure 4).

CT scans were performed for each mouse group to observe body fat distribution after the completion of spore administration. As a result, in Figure 5A, the HC and HF diets had more fat accumulation in the thorax compared to the general diet, but the fat accumulation was significantly reduced in the HC and HF diet groups administered with *B. coagulans* CC spores. Due to spore administration, the total fat pixel size in the thorax CT image was reduced compared to the non-administered group, showing a reduction rate of more than 50% in both diets.

In Figure 5B, the administration of *B. coagulans* CC spores effectively suppressed abdominal fat accumulation by HC and HF diets. Due to spore administration, the total fat pixel size in the abdominal CT image was reduced compared to the non-administered group, showing a reduction rate of more than 50% in the HC diet and more than 30% in the HF diet.

### 3.4. Feed Intake in Mice

The feed intake of the experimental animals was measured once a week during the diet experiment. Feed intake was higher in the standard diet group than in the high-carbohydrate and high-fat diet groups, and there was no significant difference in feed intake between the groups orally administered with spores and saline. Feed intake by diet was 2.74 g per day in the standard diet group, 2.05~2.08 g in the HC diet group, and 2.11~2.12 g per day in the HF diet group. In terms of feed efficiency ratio, the high-calorie diet group is higher than the standard diet group.

## 4. Discussion

*B. coagulans* is a facultative anaerobe that grows in both aerobic and anaerobic environments. It can be used as a useful gut microbe and has been reported as edible beneficial bacteria that can prevent intestinal diseases or improve immunity [37] and is used in various foods [38,39,40]. Also, as a type of lactic acid bacteria, it produces lactic acid during growth and maintains a low pH in the intestinal environment to create an intestinal environment conducive to the growth of beneficial bacteria such as lactic acid bacteria [41,42]. In addition, unlike general lactic acid bacteria, *B. coagulans* belongs to the genus *Bacillus* and forms acid-resistant spores [43], which can reach the intestine at a higher rate than other lactic acid bacteria, so its uses as gut microbes are increasing [41,44]. This study presents a new type of useful substance supply method that is directly produced in the intestine, the actual point of action of AGI, by administering spores of useful strains. AGI produced in the intestine by administering spores of the useful strain actually suppressed the absorption of monosaccharides in a high-calorie diet, thereby reducing the rate of weight gain. Unlike oral administration of pharmaceuticals, this method continuously generates AGI in the intestine, so it is expected that the effect will be improved by maintaining an appropriate, effective concentration in the body. 

DNJ has been reported to inhibit fat accumulation by inducing fat oxidation in the body [35,36,45]. In vivo, DNJ reduced adipocyte size, hepatic lipid accumulation, and plasma triacylglycerol while increasing adiponectin and activating β-oxidation to degrade fatty acids [46]. In addition, DNJ-fed obese mice lost weight, activated β-oxidation, regulated liver lipid content and adipose tissue, and showed changes in liver antioxidant enzymes and glucose metabolism; this is analyzed that it affects obesity by suppressing hyperglycemia and free radical-related stress, and consequently suppresses lipid accumulation [47]. Also, there has been a report that administration of DNJ relieved the hypertriglycemia caused by an increase in very low-density lipoprotein (VLDL) due to poor conversion of VLDL to low-density lipoprotein (LDL) in the blood. There was a report that DNJ-rich mulberry leaf extract (12 mg) was administered to nine obese adults (initial blood triglyceride level ≥200 μg) three times a day before meals for 12 weeks, reducing the main cause of arteriosclerosis, serum VLDL and increasing LDL [48]. Another study reported that 4 μM of DNJ inhibits porcine adipocyte differentiation and thereby inhibits adipogenesis. This is because DNJ inhibits the phosphorylation of extracellular regulated protein kinases 1/2 (ERK 1/2) [21]. It can be explained that it will help suppress fat accumulation by responding more sensitively to various physiological activities involved in fat metabolism as well as β-oxidation. These characteristics suggest that the weight loss and body fat accumulation inhibitory effects of DNJ reported in this study can be partly explained by the glucose absorption inhibitory and body fat oxidation promoting effects. According to the results in Figure 5, the administration of *B. coagulans* CC spores significantly suppressed fat accumulation in a high-calorie diet, and this is judged to be due to the physiological effects of DNJ discussed above [21,35,36,45,46,47,48]. 

On the other hand, a purification process is required to process physiologically active substances from natural products such as fermented products or plants [49,50]. Purification of the extract is performed for the purpose of concentration of active ingredients, removal of impurities, decolorization, and deodorization, and it is a cumbersome process that not only takes time and money but also requires yield improvement [51]. To solve this inconvenience in the purification process, the results of this study suggest that a method of producing and supplying physiologically active substances in the body by orally administering a specific strain that easily reaches and grows in the intestine can be a countermeasure.

## 5. Conclusions

In this study, we examined the possibility of action in the body by producing and supplying useful substances in the intestine. As a result, the possibility of a new production method for producing specific useful substances in the intestine by orally administering spores of anaerobe was suggested. Bioactive substances that can be produced by edible bacteria are limited but have various application potentials. Since the *B. coagulans* CC strain does not settle in the intestine for a long time and is excreted from the body within a few weeks, it has the advantage of being usable as a useful substance-producing bacteria within a limited period. In this study, it was proven that the effects of AGI produced in the intestine and industrially produced were the same. Extending this model, studies are being conducted on bioactive substances other than AGI that require effective blood level maintenance.

## Figures and Tables

**Figure 1 microorganisms-11-01462-f001:**
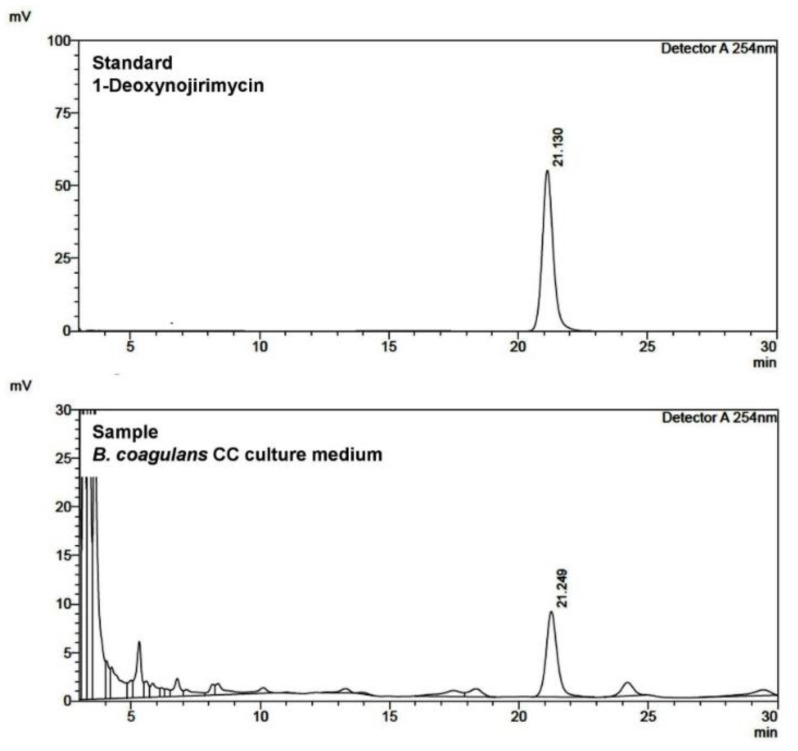
HPLC analysis of *B. coagulans* CC culture. The standard product, 1-deoxynojirimycin, and the sample are detected at the same retention time.

**Figure 2 microorganisms-11-01462-f002:**
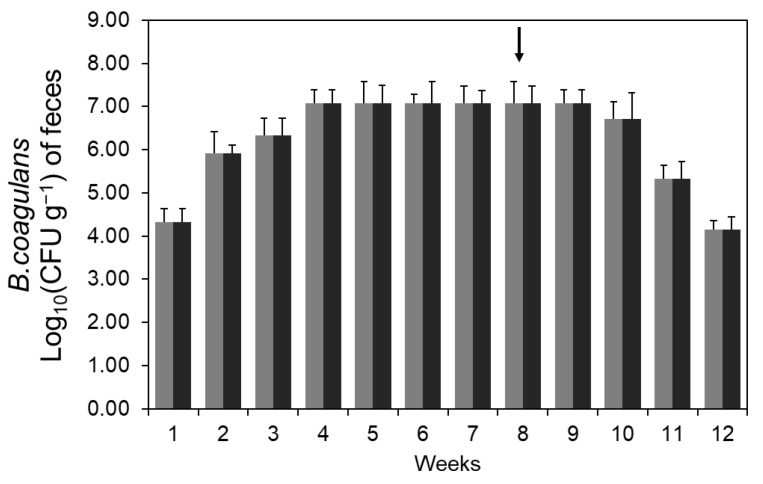
The number of *B. coagulans* CC excreted in feces. Three-week-old ICR mice were classified into eight animals per group and four animals per cage. During the breeding period, feces were collected and measured the number of microorganisms contained in the feces at regular intervals. After three weeks of intake, the spore-administered group showed a constant number of microorganisms. Each symbol means an HC diet with *B. coagulans* CC spores (
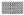
) and an HF diet with *B. coagulans* CC spores (

). The arrow mark means the cessation of spore administration.

**Figure 3 microorganisms-11-01462-f003:**
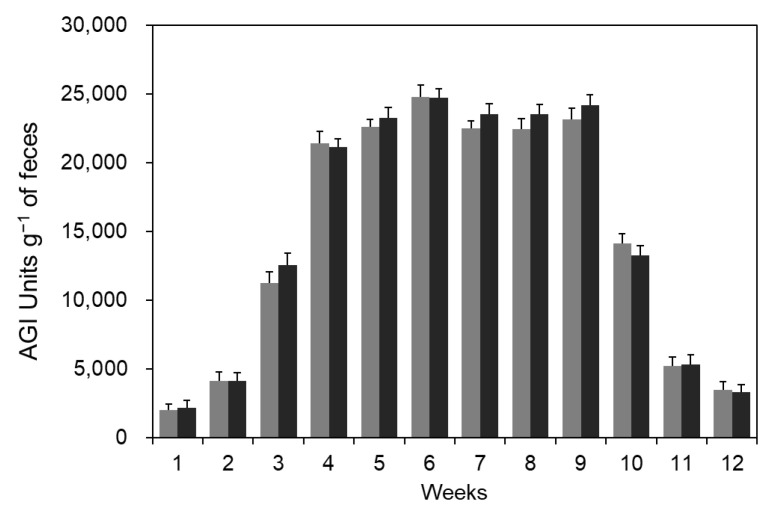
The amount of AGI excreted in feces. Three-week-old ICR mice were classified into eight animals per group and four animals per one cage. During the breeding period, feces were collected and analyzed the amount of AGI contained in the feces at regular intervals. After three weeks of intake, the spore-administered group showed a constant amount of AGI. Each symbol means an HC diet with *B. coagulans* CC spores (
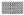
) and an HF diet with *B. coagulans* CC spores (

). The arrow mark means the cessation of the diet point of *B. coagulans* CC spores.

**Figure 4 microorganisms-11-01462-f004:**
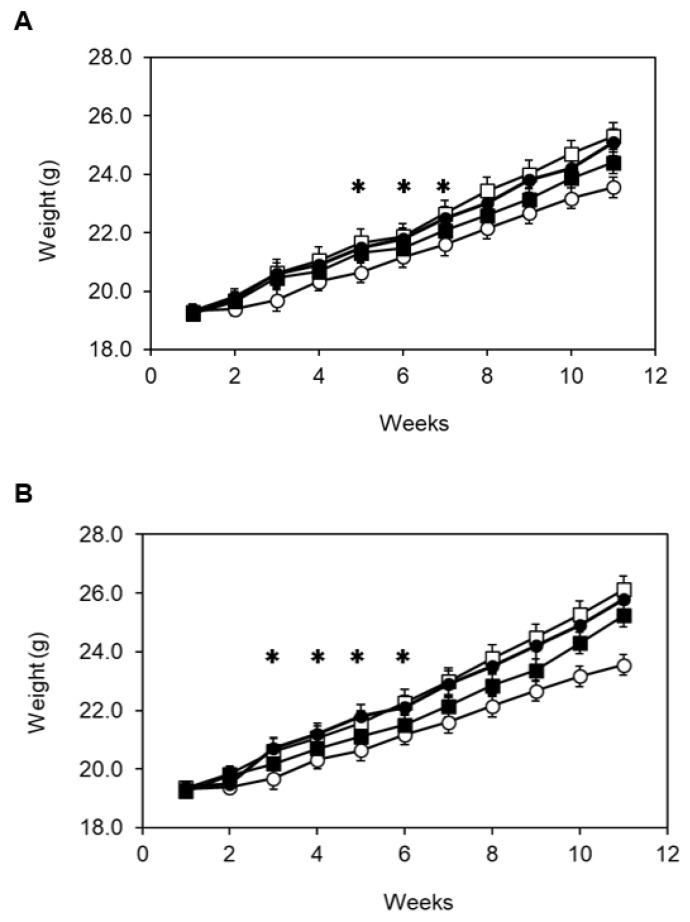
Effect of administration of spores of *B. coagulans* CC on weight gain in the high-calorie diet. Three-week-old ICR mice were classified into eight animals per group and four animals per one cage. During the breeding period, the weight-gaining patterns with and without spore administration were compared in the high-carbohydrate (HC, A) and high-fat (HF, B) diet groups. (**A**) Each symbol means an HC diet (□), an HC diet with *B. coagulans* CC spores (■), a standard diet (○), and an HC diet with *B. coagulans* ATCC7050 spores (●). (**B**) Each symbol means an HF diet (□), an HF diet with *B. coagulans* CC spores (■), a standard diet (○), and an HF diet with *B. coagulans* ATCC7050 spores (●). Values were considered to be significant (✱) when *p* was less than 0.05 (*p* ≤ 0.05). ✱ means that the high-calorie diet group with and without *B. coagulans* CC was shown to be accompanied by reliability, meaning significance.

**Figure 5 microorganisms-11-01462-f005:**
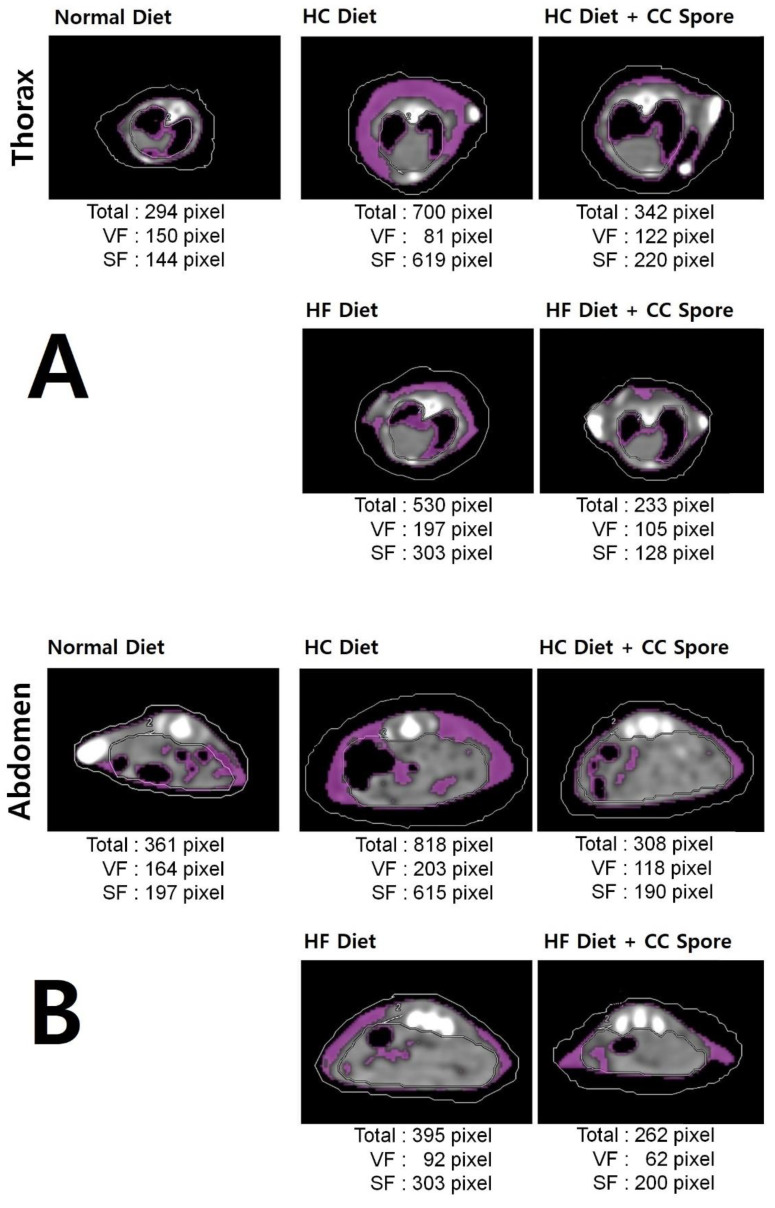
Validation of changes in fat accumulation by computed tomography. After 7 weeks of *B. coagulans* CC spore administration, animals from the normal diet group, high-carbohydrate diet group (HC diet), and high-carbohydrate diet group with spores (HC diet + CC spore), high-fat diet group (HF diet), and high-fat diet group with spores (HF diet + CC spore) were selected, anesthetized, and (**A**) thorax and (**B**) abdominal of transverse section computed tomography (CT) was taken. The CT scan of each group showed the area of visceral fat (VF) and subcutaneous fat (SF) in pixel units, and the purple area indicated a fat area.

**Table 1 microorganisms-11-01462-t001:** Comparison of AGI activity of various strains in aerobic, anaerobic cultures.

Strains	Incubation Time (Hours)	AGI (Unit mL^−1^)	
		Aerobic	Anaerobic
*B. subtilis* DC-15	24	83,257	N.D. ^a^
*B. coagulans* ATCC7050	48	N.D. ^a^	N.D. ^a^
*B. coagulans* CC	48	19,478	12,879

^a^ N.D.; Not detected.

## Data Availability

All data generated or analyzed during this study are included in this article.

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
