# Peer review of "Intestinal Production of Alpha-Glucosidase Inhibitor by Bacillus coagulans Spores"

_microorganisms, 2023, doi:10.3390/microorganisms11061462_

Round 1
Reviewer 1 Report
This manuscript isolated Bacillus coagulans CC with high AGI-producing activity and excellent sporulation ability. Administrated Bacillus coagulans CC to mice for a few weeks, the high-carbohydrate diet and the high-fat diet showed a 5% lower weight gain compared to the non-administrated group. Based on computed tomography, the visceral and subcutaneous fat layers of the abdomen and thorax in both high-carbohydrate and high-fat diet groups compared to the non-administered group were decreased. The results are promising, major revision need.
Comments:
1. Only 2 pictures and one table, the number of figures is small and a lot of data do not show. Please provide it. Such as L172, L198 and L203.
2. Do you have experiment replicates for Figure 1 and Figure 2, how about the result? Similar?
3. In Figure 2, please provide and place the pixel unit as a figure, add biological replicates.
4. In L187, why do you pick up yellow colonies?
Author Response
- Thank you for kind review.
- The revised manuscript is attached.
- Only 2 pictures and one table, the number of figures is small and a lot of data do not show. Please provide it. Such as L172, L198 and L203.
-> HPLC data was added in L179. Data corresponding to L198 and L203 were added in L204 and 224 respectively. It showed a similar trend to our previously reported spore administration paper, which is described in L219. Please note that some of the data have been reduced due to the large amount of the manuscript, as this manuscript covers everything from bacterial selection to in vivo experiments.
- Do you have experiment replicates for Figure 1 and Figure 2, how about the result? Similar?
-> As the first experiment, a spore administration experiment was conducted and confirmed, and then the main experiment was conducted. In the first experiment, only weight change and detection of bacteria and AGI in feces were carried out, and the same results were confirmed as in this experiment.
- In Figure 2, please provide and place the pixel unit as a figure, add biological replicates.
-> The pixel units are listed in Figure 5. CT scan was performed by selecting one representative animal for each group. The variation between subjects in each group can be replaced by the SD bar of the weight graph in Figure 4.
- In L187, why do you pick up yellow colonies?
-> Bromocresol purple, which is contained in the BCP medium used for colony selection, is a pH indicator and turns yellow in acid. Since yellow colony means that it produces acid, we selected yellow colony to screen for lactic acid-producing bacteria such as Bacillus coagulans. In a culture environment of 55℃, yellow colonies were not found in the feces of the group not administered with Bacillus coagulans spores, and yellow colonies were detected only in the feces of the group administered with Bacillus coagulans spores. Therefore, it was judged as Bacillus coagulans.

Reviewer 2 Report
The authors discovered that Bacillus coagulans CC, a facultative anaerobe isolated from hay, produces a significant amount of α-glucosidase inhibitor, identified as 1-deoxynojirimycin. The inhibitor activity was confirmed in mice intestines and feces after oral administration of the bacteria's spores. The study showed that Bacillus coagulans CC could efficiently reach the intestines, proliferate, and produce α-glucosidase inhibitors. Mice given the spores experienced a 5% lower weight gain from high-carbohydrate and high-fat diets and reduced fat layers in their abdomen and thorax compared to the non-administered group. This indicates that the α-glucosidase inhibitor produced by this specific strain can work efficiently in the intestine. This study is intriguing and well-written, with clear logic and results that strongly support the conclusions. The authors are advised to consider the following suggestions for revisions:
- The authors mention in the introduction that previous studies have been conducted to test the efficacy of AGIs derived from natural products. However, these natural products have many practical limitations, such as reduced palatability due to smell and color. It seems that the current study has not investigated these aspects of reduced palatability, such as smell and color.
- The authors employed 3-week-old ICR mice for the experiments. It is requested that the authors provide a rationale for using animals of this specific age.
- In lines 196-199, the authors state that after ceasing spore administration at the 8th week of the diet, the spores were continuously detected until the 2nd week and then gradually decreased, suggesting that the intestinal residence time of this strain is no more than several weeks. However, the authors do not provide the corresponding data. To greatly enhance the completeness and scientific rigor of this study, the reviewer recommends including this data in the manuscript. Additionally, it is recommended that the authors engage in a more in-depth discussion within the manuscript's discussion section.
- A more comprehensive discussion of the study's limitations is warranted to provide a balanced perspective on the findings.
Author Response
- Thank you for kind review.
- The revised manuscript is attached.
- The authors mention in the introduction that previous studies have been conducted to test the efficacy of AGIs derived from natural products. However, these natural products have many practical limitations, such as reduced palatability due to smell and color. It seems that the current study has not investigated these aspects of reduced palatability, such as smell and color.
-> AGI produced as a secondary metabolite by plants or bacteria has its own color and flavor, so palatability should be considered when ingesting it. However, the AGI intake method used in this experiment was produced in the intestine by administering the bacterial spores. Compared to existing natural product-derived AGI, intake is significantly less, and it does not show a distinctive odor or color, so there is no need to consider palatability.
- The authors employed 3-week-old ICR mice for the experiments. It is requested that the authors provide a rationale for using animals of this specific age.
-> 3-week-old mice were selected because they are in the growth phase and metabolic activities are active. The diet experiment was conducted after a 1-week adaptation period and was conducted at 4 weeks of age.
- In lines 196-199, the authors state that after ceasing spore administration at the 8th week of the diet, the spores were continuously detected until the 2nd week and then gradually decreased, suggesting that the intestinal residence time of this strain is no more than several weeks. However, the authors do not provide the corresponding data. To greatly enhance the completeness and scientific rigor of this study, the reviewer recommends including this data in the manuscript. Additionally, it is recommended that the authors engage in a more in-depth discussion within the manuscript's discussion section.
-> Corresponding data is added to line 204. Please note that some of the data have been reduced due to the large amount of the manuscript, as this manuscript covers everything from bacterial selection to in vivo experiments.
- A more comprehensive discussion of the study's limitations is warranted to provide a balanced perspective on the findings.
-> The limitations of the study were mentioned in the discussion part because the types of bacteria that can be eaten are limited. Since it is a new production method of intestinal production and direct supply of physiologically active substances by bacteria grown in the intestine, it is judged that there is a possibility of research on the intestinal production of various physiologically active substances. In addition, in this paper, the production and action point of AGI coincided with the intestine, so the spread of bioactive substances produced in the intestine to various places was not confirmed. Additional studies are now confirming that some of physiologically bioactive substances produced in the intestine flow into the bloodstream.

Reviewer 3 Report
he findings are promising for alternative glycemic-lowering agents. The authors attempted to prove the capability of Bacillus coagulans spores in reducing fat accumulation regarding increased Alpha-glucosidase inhibitor production.
However, I have some comments.
Major revisions:
1. The mentioned "Strains" lack an origin, is it from commercial or any special harvest? (Lines 69-76). Please provide detailed information.
2. The formular of inhibition rate was acquired by the author group or any relevant guidance? (Line 88).
3. Please give the detail information of the CT scan for taking the picture of VF and SF.
4. In the animal model, a high-fat diet was given to mice as 26.1% of fat. Although the exact percentage of fat content can vary depending on the specific research goals and the animal model being used, usually, in research studies using animal models, a high-fat diet is typically defined as containing 40% or more of calories from fat. What is the relevant explanation for the 26.1% fat content in the present study? (Lines 118-139).
5. In Figure 2B, authors subjectively present individual subjects' results, not the average value of VF and SF derivedfrom certain groups. Figures should be provided to compare different fat masses (VF and SF) in different diets and CC Spores. Please provide the mean/median and standard deviation and statistical analysis.
6. According to increased AGI caused by B. coagulans CC spore, it should be explained carefully how AGI could reduce the much higher mass of VF and SF among mice fed by HF diet than those fed by HC diet.
Author Response
- Thank you for kind review.
- The revised manuscript is attached.
- The mentioned "Strains" lack an origin, is it from commercial or any special harvest? (Lines 69-76). Please provide detailed information.
-> Bacillus coagulans CC was isolated from domestically collected hay and silage. t is described on line 70.
- The formular of inhibition rate was acquired by the author group or any relevant guidance? (Line 88).
-> This formula is referenced in reference 30 and is described at line 85.
- Please give the detail information of the CT scan for taking the picture of VF and SF.
-> The specifications of the CT scanner are described on line 139.
- In the animal model, a high-fat diet was given to mice as 26.1% of fat. Although the exact percentage of fat content can vary depending on the specific research goals and the animal model being used, usually, in research studies using animal models, a high-fat diet is typically defined as containing 40% or more of calories from fat. What is the relevant explanation for the 26.1% fat content in the present study? (Lines 118-139).
-> We made and used high-fat feed ourselves, and I know that it is common to use 20-40% of the daily dietary amount as fat. It is described as a content ratio in the Manuscript, and when calorie is calculated, the fat calorie is about 50% of the total calorie of the high-fat feed.
- In Figure 2B, authors subjectively present individual subjects' results, not the average value of VF and SF derivedfrom certain groups. Figures should be provided to compare different fat masses (VF and SF) in different diets and CC Spores. Please provide the mean/median and standard deviation and statistical analysis.
-> CT scan was performed by selecting one representative animal for each group. The variation between subjects in each group can be replaced by the SD bar of the weight graph in Figure 4.
- According to increased AGI caused by B. coagulans CC spore, it should be explained carefully how AGI could reduce the much higher mass of VF and SF among mice fed by HF diet than those fed by HC diet.
-> Line 293 explains the β-oxidation of fat activated by DNJ, the main component of AGI. In addition, there has been a report that administration of DNJ relieved hypertriglycemia caused by an increase in VLDL due to poor conversion of VLDL to LDL in the blood. There was a report that serum VLDL, the main cause of arteriosclerosis, decreased and LDL increased when DNJ-rich mulberry leaf extract (12mg) was administered three times a day before meals to 9 obese adults (initial blood triglyceride level ≥200ug) for 12 weeks. In another study, it was reported that 4 μM of DNJ inhibited adipogenesis by suppressing the differentiation of porcine adipocytes. This is because DNJ inhibits the phosphorylation of extracellular regulated protein kinases 1/2 (ERK1/2). It can be explained that it will help suppress fat accumulation by responding more sensitively to various physiological activities involved in fat metabolism as well as β-oxidation (Line 321~331).

Reviewer 4 Report
Dear Editor,
I have reviewed the manuscript, microorganisms-2302337.
Unfortunately, I cannot recommend the manuscript be published in Microorganisms.
English and text need to be significantly improved. Some, but not all, comments can be found below.
Comments:
- Citations in the text need to be improved: lines 35, 36.
- Italic font must be applied for Latin text: in vivo (line 63).
- What is the meaning of the abbreviations in line 74?
- Please specify the producer of soy flour medium (line 77).
- Incomplete sentence in line 88.
- Please specify the producer of HPLC equipment.
- Please specify the producers and purity of the used chemicals (lines 112-113).
- A reference to the software for the statistical analysis is missing.
- Table 1—wrong text format.
- Figure 1 needs to be formatted according to the Instructions for authors.
- The Discussion subchapter is weak.
- Conclusions must be moved to a separate chapter.
Author Response
- Thank you for kind review.
- The revised manuscript is attached.
1. Citations in the text need to be improved: lines 35, 36.
-> We have changed that part. It seems to be missing as the journal publisher changed the word file format.
2. Italic font must be applied for Latin text: in vivo (line 63).
-> We have changed that part.
3. What is the meaning of the abbreviations in line 74?
-> It is the medium reagent abbreviation, manufacturer, and product number.
4. Please specify the producer of soy flour medium (line 77).
-> The soybean flour medium was made by us with soybeans. The fabrication method is described on line 79.
5. Incomplete sentence in line 88.
-> It is the AGI inhibition rate calculation formula.
6. Please specify the producer of HPLC equipment.
-> It is attached to Line 94.
7. Please specify the producers and purity of the used chemicals (lines 112-113).
-> Mineral reagents used in the sporulation medium were specially sold reagents, and those with guaranteed purity were used.
8. A reference to the software for the statistical analysis is missing.
-> It is described on Line 142.
9. Table 1—wrong text format.
-> The font style of table is free style. The font of the table has been unified with the font style of the body text. Also, paragraphs are separated to distinguish between the main text and tables.
10. Figure 1 needs to be formatted according to the Instructions for authors.
-> Since the figure caption and the manuscript could not be distinguished, the paragraph following the figure caption was separated. This happened when the journal publisher changed the word document format.
11. The Discussion subchapter is weak.
-> In the discussion, the β-oxidation of fat activated by DNJ, the main component of AGI, was explained. In addition, there has been a report that administration of DNJ relieved hypertriglycemia caused by an increase in VLDL due to poor conversion of VLDL to LDL in the blood. There was a report that serum VLDL, the main cause of arteriosclerosis, decreased and LDL increased when DNJ-rich mulberry leaf extract (12mg) was administered three times a day before meals to 9 obese adults (initial blood triglyceride level ≥200ug) for 12 weeks. In another study, it was reported that 4 μM of DNJ inhibited adipogenesis by suppressing the differentiation of porcine adipocytes. This is because DNJ inhibits the phosphorylation of extracellular regulated protein kinases 1/2 (ERK1/2). It can be explained that it will help suppress fat accumulation by responding more sensitively to various physiological activities involved in fat metabolism as well as β-oxidation (Line 321~331).
12. Conclusions must be moved to a separate chapter.
-> Conclusions section was separated on line 346.

Round 2
Reviewer 2 Report
The authors have revised their manuscript, which is now suitable for publication.Reviewer 4 Report
The manuscript still needs attention to improve the technical side and style.
The Discussion and Conclusions subchapters need to be improved.
Please read the Instructions for authors and the comments of reviewers.